# Massive Axial and Appendicular Skeletal Deformities in Connection with Gorham-Stout Syndrome

**DOI:** 10.3390/medicines6020054

**Published:** 2019-05-07

**Authors:** Ali Al Kaissi, Sami Bouchoucha, Mohammad Shboul, Vladimir Kenis, Franz Grill, Rudolf Ganger, Susanne Gerit Kircher

**Affiliations:** 1Ludwig Boltzmann Institute of Osteology, at the Hanusch Hospital of WGKK and, AUVA Trauma Centre Meidling, First Medical Department, Hanusch Hospital, 1140 Vienna, Austria; 2Orthopaedic Hospital of Speising, Paediatric Department, 1130 Vienna, Austria; grill.franz@gmx.net (F.G.); rudolf.ganger@oss.at (R.G.); 3Paediatric Orthopedic Surgery—Children Hospital, Tunis 1029, Tunis-Tunisia; sami.bouchoucha@yahoo.com; 4Department of Medical Laboratory Sciences, Jordan University of Science and Technology, Irbid 22110, Jordan; mohammad.shboul@reversade.com; 5Department of Foot and Ankle Surgery, Neuroorthopaedics and Systemic Disorders, Pediatric Orthopedic Institute n.a. H. Turner, Parkovaya str., 64-68, 196603 Pushkin, Saint Petersburg, Russia; kenis@mail.ru; 6Department of Medical Chemistry, Medical University of Vienna, 1090 Vienna, Austria; susanne.kircher@meduniwien.ac.at

**Keywords:** spine deformities, joints and long bone deformities, vanishing bone disorder, gorham-stout disease, CT scan

## Abstract

**Background**: Etiological understanding is the corner stone in the management of skeletal deformities. **Methods**: Multi-centre study of patients with deformities in connection with diverse etiological backgrounds. We aimed to study four patients (one boy and three girls) with variable axial and appendicular deformities in connection with a vanishing bone disorder. **Results**: Axial deformities such as scoliosis, kyphoscoliosis, compressed fused vertebrae, appendicular fractures, dislocations, and vicious disorganization deformities of the joints were in connection with the vanishing bone disorder, namely Gorham-Stout syndrome. **Conclusions**: It is mandatory to establish proper clinical and radiological phenotypic characterization in children and adults presented with unusual skeletal deformities. Identifying the reason behind these deformities is the key factor to draw a comprehensive management plan.

## 1. Introduction

Earlier reports (Jackson et al. [1] and Gorham and Stout [2]) described massive osteolysis or vanishing bone disease as a rare and enigmatic condition involving various skeletal locations. Gorham–Stout disease (GSD; OMIM 123880) also known as vanishing bone disease, phantom bone disease, massive osteolysis, Gorham–Stout syndrome, and Gorham’s disease is a rare entity characterized by destruction of osseous matrix and proliferation of vascular structures with benign origin [3]. Despite the extensive investigation of the pathogenetic mechanisms of the disease, its etiology of GSD is poorly understood and the cause of excessive bone resorption in GSD patients has not been clarified. Several studies proposed that the abnormal proliferation of endothelial-lined vessels could promote bone resorption [4]. Bone normally do not have lymphatic vessels and the presence of lymphatics in GSD patients have been confirmed histologically and biochemically. Wang et al. [5] has recently demonstrated that the elevated level of the macrophage colony-stimulating factor (M-CSF) in mice given intra-tibial lymphatic endothelial cells (LECs) stimulated osteoclasts formation and activation. Moreover, several markers such as LYVE-1 and podoplanin VEGF-A, VEGF-C, and PDGFRβ in addition to interleukin-6 were increased in serum and vessels of GSD patients; all of which have the ability to increase osteoclast activity [6,7,8,9,10].

Different classifications, based on the genetic transmission of the osteolysis in particular, have been proposed. Studies by Hardegger et al. [11] revealed the most commonly accepted classification; type 1, hereditary multicentric osteolysis with dominant transmission; type 2, hereditary multicentric osteolysis with recessive transmission, type 3, nonhereditary multicentric nosteolysis with nephropathy; type 4, Gorham-Stout syndrome; and type 5, Winchester syndrome defined as a monocentric disease of autosomal recessive inheritance. 

Gorham-Stout disease can affect any part of the skeleton, but the pelvis, long bones, and shoulder girdles are the most frequently involved [12,13,14]. Chylous pericardial and pleural effusions may occur due to mediastinal extension of the disease process from the involved vertebra, scapula, rib or sternum, and can be life threatening. A high morbidity and mortality is seen in patients with spinal and/or visceral involvement [15,16]. The diagnosis of GD can be made based on clinical, imaging, and histological findings, although the clinical and the radiological phenotypes were sufficient to establish the diagnosis in most cases.

## 2. Materials and Methods 

### 2.1. Participants and Ethics

The study protocol was approved and obtained from the Hospital Ethical Committee (Ethics Committee of the Turner Scientific Research Institute, No.3/2016, Saint-Petersburg, approval date: 5 January 2016), in addition informed consent was obtained from the patient’s guardians. The records of four patients (one boy and three girls) of different ethnic origins were studied at the osteogenetic department of the Orthopaedic Hospital of Speising, Vienna.

All our patients showed normal course of development and their family histories were non-contributory. Clinical presentations were variable and strongly correlated to the site of the skeletal involvement. However, intractable pain associated with swellings of the soft tissues, was a uniform presentation in all our patients, without any reason behind. 

### 2.2. Clinical Examination

Clinical examination, showed all our patients were of normal height, without any associated facial dysmorphic features or abnormal craniofacial contour. Neurological, vision, and hearing examinations were normal. All had received painkillers by their physicians to alleviate intractable pain. Severe limitations of daily activities have been observed. Patients with involvement of the spine showed severe and acute restrictions of the physiological spine biomechanics associated with progressive deformities, which ranged between painful kyphosis to kyphoscoliosis. Patients who manifested early life bone fractures after minor trauma showed massive painful involvement of the whole limb. The extension of the pathology to involve the weight bearing zones was a catastrophic outcome of the pathology. Karyotypes were normal.

### 2.3. Laboratory Measurements

Laboratory investigations such as serum and urinary oligosaccharides, mucopolysaccharides, serum lactate, pyruvate, creatine phosphokinase, calcium, full blood count, serum crosslaps, parathyroid hormone, 25-hydroxy vitamin D, Erythrocyte sedimentation rate (ESR), antinuclear antibody (ANA), and rheumatoid factors (FR) were measured for patients. 

## 3. Results

We divided our patients into two groups in accordance with the clinical presentation:

### 3.1. Group I: Patients with Progressive Painful Tilting of the Spine 

A seven-year-old-boy was seen for the first time at the age of three years because of painful thoraco-lumbar kyphosis resulted from osteolysis of T9-L1. The left iliac crest was also involved in the pathological process of osteolysis (Figure 1). Back pain associated with Trendelenburg gait were the most bothersome symptomatology. Bone biopsy confirmed the diagnosis of Gorham-Stout disease. The kyphosis has been treated by a brace. Follow-ups showed stability of the kyphosis and dormancy of the osteolysis.A ten-year-old-girl presented with painful acute upper thoracic kyphosis. Radiographs of the spine were difficult to assess. 3D reformatted CT scan of the thoracic spine showed progressive osteolysis of T3-T6 (Figure 2). Skeletal survey did not show any other involved areas.3D reconstruction CT scan in a-20-years-old-girl showed severe flattening, fusion, shrinkage and compression of the vanished thoracic spine T3-T9 causing effectively the development of painful kyphoscoliosis. Vanishing bone extended to involve the right shoulder joint resulted in total drop of the right upper limb (Figure 3).

### 3.2. Group II: Patients with Progressive Disorganization of the Weight Bearing Joints Associated with Painful Dissolution of the Long Bones

A-13-year-old-girl was seen for the first time at the age of 6-years because of fracture of the left tibia. The fracture was treated via open reduction with plate fixation. Sadly speaking, at the age of 13-years she presented with Trendelenburg gait because of vicious osteolysis, which involved the entire lower left limb, which resulted in marked limb length discrepancy. At the age of 17-years, the osteolysis spread along the whole right lower limb. The hips and the left clavicle were also involved. The osteolysis extended over the entire left inferior limb and an important limb length discrepancy and deformity were the outcome. Over a period of several years the osteolysis extended to the opposite limb, both hips were involved. Anteroposterior pelvis radiograph showed severe osteolytic involvement of the hips associated with massive disorganization/distortion of the weight bearing components with subsequent development of lower limb length inequality. The patient was able to walk a few days after surgery using crutches with partial weight bearing. Osteogenesis imperfecta was the first proposed diagnosis because of long bone fracture and osteopenia. Next-generation sequencing panel to detect *COL1A1/2* mutations was negative (Figure 4). A bone specimen of a lesion over the left femur obtained at an open biopsy. The histopathology of the bone biopsy via original magnification revealed trabecular fragments and large cavernous lymph spaces (Figure 4). A bone specimen of a lesion over the left femur obtained at an open biopsy. Overview bone biopsy (H&E staining): (**a**) Via original magnification, the histopathology revealed trabecular fragments and large cavernous lymph spaces. (**b**) Another bone biopsy at higher magnification showed lymphatic malformation with ectatic lymph vessels. a. magnification of ×100. b. Higher magnification (×400) (Figure 5).

Serum and urinary oligosaccharides, mucopolysaccharides, serum lactate, pyruvate, creatine phosphokinase, calcium, full blood count, in addition to serum crosslaps and parathyroid hormone parameters were within the normal values. Serum 25-hydroxy vitamin D was estimated by radioimmunoassay to be 41 ng/mL (normal levels are 36–48 ng/mL). Erythrocyte sedimentation rate (ESR) was unremarkable and ranged between 10–15 mm/1st hour. Antinuclear antibody (ANA) and rheumatoid factors (FR) were negative. Two adult patients showed slight proteinuria and raised alkaline phosphatase (reflecting active bone turnover).

## 4. Discussion

Gorham–Stout disease (GSD, or the so called massive osteolysis, vanishing bone disease, and or phantom bone disease. GSD is characterised by abnormal proliferation of non-neoplastic vascular and lymphatic tissue (angiomatous proliferation) and at a later stage, by fibrous tissue resulting in a massive destruction and resorption of affected bones, which lead to skeletal deformities and functional impairment often associated with swelling [17]. GSD usually diagnosed in children and young adults but it can manifest at any age (between 18 months and 60 years) and is not restricted to gender or race [1,2]. The presenting symptom is usually pain in a long bone, the pelvis, thorax or spine. Routine radiological examinations such as X-rays, bone scan, computed tomography (CT) and magnetic resonance imaging (MRI) are useful for the diagnosis. The disease can also be confirmed by histopathological alteration in bone biopsy from the lytic bone in addition to other features distinguishing Gorham’s disease from other conditions associated with bon destruction as suggested by Heffez and colleagues [18]. The causation of GSD is still speculative, the aetiology and pathophysiology are also still largely unknown, and there is no apparent genetic predisposition with an unpredictable course and progression. More than 300 cases of GSD have been described in the literature [4]. Several studies have suggested the involvement of RANK signalling [4,19] or other biomarkers such as platelet-derived growth factor (PDGF), interleukin-6, and vascular endothelial growth factor (VEGF) [4,6,9]. However, these biomarkers have the ability to increase osteoclast function, they are neither indicative of the diagnosis nor elevated in all GSD patients. Wang et al. [5] has recently demonstrated that lymphatic endothelial cells (LECs)- injected mice express high levels of macrophage colony-stimulating factor (M-CSF); a factor that stimulates osteoclasts formation. LECs-injected mice displayed sever bone loss and destruction and increase of osteoclasts activity; phenotype similar to those of GSD patients such as osteoclastogenesis, massive osteolysis, and bone destruction. Osteolytic phenotype in mice were blocked by an M-CSF neutralizing antibody and significantly diminished tibial bone destruction.

The clinical presentations are variable, largely depending upon the site of skeletal involvement and presence of systemic manifestations. A single or multiple bone can be affected and it may involve any part of the skeleton. Because of the rarity of GSD, the diagnosis of this syndrome is challenging for clinicians and requires the exclusion of other osteolysis-associated disorders such as inflammatory, metabolic, infectious (e.g., osteomyelitis), malignant neoplasm (e.g., squamous cell carcinoma), other hereditary, traumatic, and endocrine diseases. Extensive metastatic bone disease due to breast cancer, and osteosarcoma as well as aneurysmal bone cyst, is some of the conditions that resemble vanishing bone disease and can be distinguished by a biopsy [20,21]. Moreover, there is no definitive treatment to treat this disorder has been established yet and most patients have been treated with surgery and or radiation therapy [22,23]. Other conventional therapies have also been tried using drugs such as interferon-α2b, which was found to effective chylothorax treatment [23] in addition to RANK-ligand inhibitor (Denosumab), anti-VEGF-A antibody (Bevacizumab), steroids, vitamin D, and calcitonin to control of progressive osteolytic activity and to slow mandibular resorption [21,24,25,26].

The disease pathophysiology commences with intramedullary and subcortical radiolucent foci resembling patchy osteoporosis. It makes slow, irregular, local progress with a concentric shrinkage of the shafts of the bones, the affected bone disappears more or less completely unless spontaneous remission occurs. Pathological fractures rarely heal and the osteolytic process continues through the fragment [13,14].

The pathological process in Gorham disease may affect the appendicular or the axial skeleton. Previous reports showed that the shoulders are the most common sites of involvement [16,27]. Other sites of involvement such as the humerus, scapula, clavicle, ribs, sternum, pelvis and femur, mandible, hand, spine, and cervical spine have been reported [12,15,28,29,30,31,32]. Involvement of the ribs, scapula, or thoracic vertebrae may lead to the development of the pleura (chylothorax) from direct extension of lymphangiectasia into the pleural cavity or via invasion of the thoracic duct [15,33]. Chylothorax is the most serious complication, that without surgical intervention can result in respiratory distress and failure and it is associated with high rate of morbidity and mortality [3,34,35].

## 5. Conclusions

There was no evidence of a malignant, neuropathic, or infectious elements involved in the causation of this disorder in our patients. There are no known definitive methods to treat this disorder but various forms of antiangiogenic invasive agents and/or drugs are under study. In addition, various forms of chemotherapeutic agents are under trial as well. Radiotherapy has been used but often proved ineffective. The aim of this study is to alert physicians and orthopaedic surgeons, the necessity to distinguish fractures and or osteolysis in connection with vanishing bone disorder from other diagnoses such as osteogenesis imperfecta. Comprehensive clinical and the radiological phenotypic characterizations are the base line tools toward proper management.

## Figures and Tables

**Figure 1 medicines-06-00054-f001:**
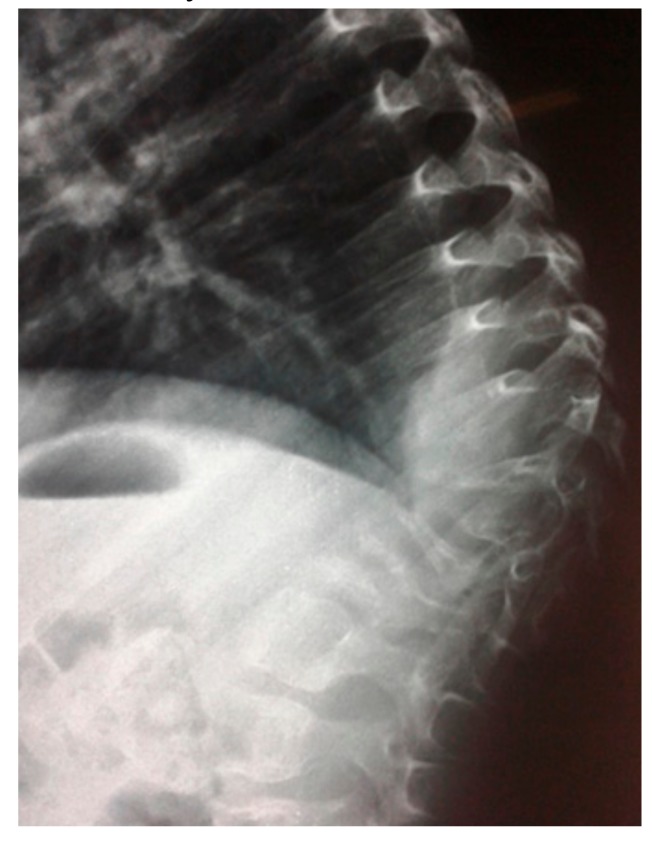
Lateral spine radiograph of a 7-year old boy showed kyphosis resulted from osteolysis of T9-L1 (fish-like vertebrae). The left iliac crest was also involved in the pathological process of osteolysis.

**Figure 2 medicines-06-00054-f002:**
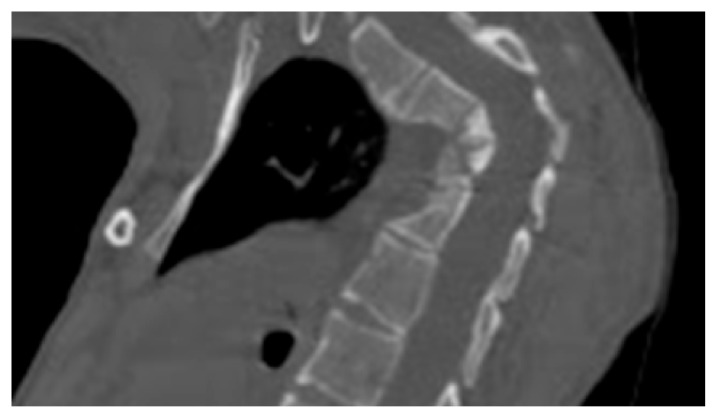
3D reformatted CT scan in a 10-years-old-girl showed progressive osteolysis of T3-T6. Resulted in the development of progressive acute kyphosis of 90° Cobbs angle. Skeletal survey did not show any other involved areas.

**Figure 3 medicines-06-00054-f003:**
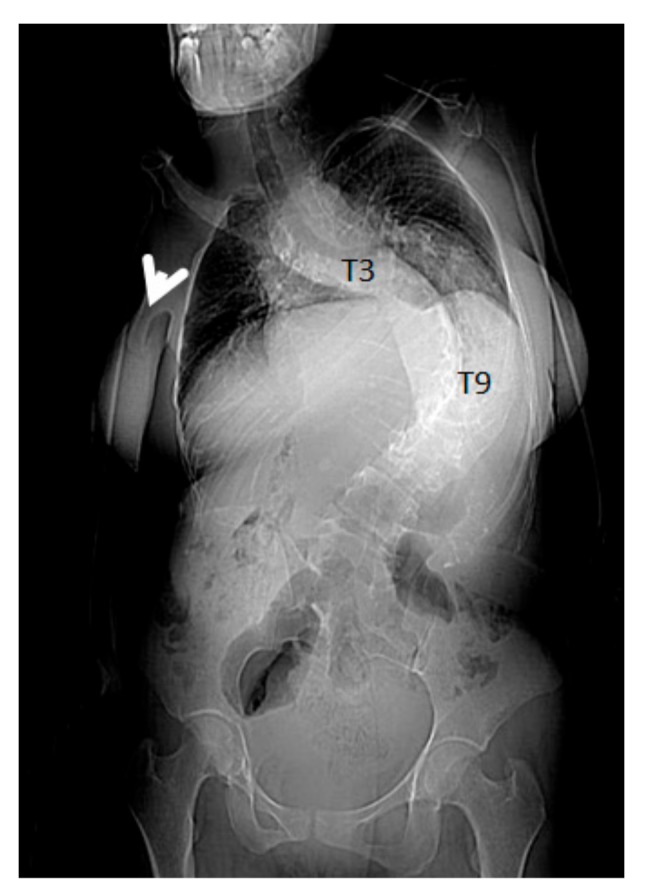
3D reconstruction CT scan in a-20-years-old-girl showed severe flattening, fusion, shrinkage and compression of the dissolved thoracic spine T3-T9. Vanishing bone extended to involve the right shoulder joint resulted in total drop of the right upper limb (arrow). The combination of vertebral body osteolysis and unilateral right shoulder joint destruction resulted in the development of progressive kyphoscoliosis of 90° Cobbs angle. Note the dislocated right clavicle.

**Figure 4 medicines-06-00054-f004:**
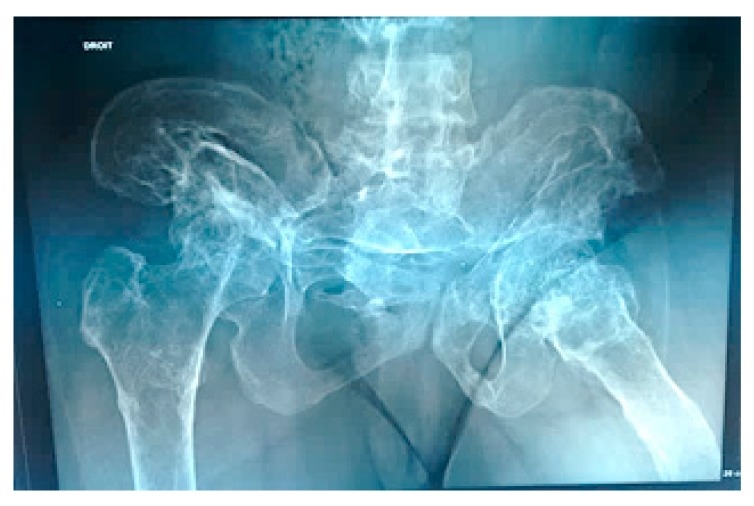
Anteroposterior pelvis radiograph showed severe osteolytic involvement of the hips associated with massive disorganization/distortion of the weight bearing components with subsequent development of lower limb length inequality. The patient was able to walk a few days after surgery using crutches with partial weight bearing. Osteogenesis imperfecta was the first proposed diagnosis because of long bone fracture and osteopenia. Next-generation sequencing panel to detect *COL1A1/2* mutations was negative.

**Figure 5 medicines-06-00054-f005:**
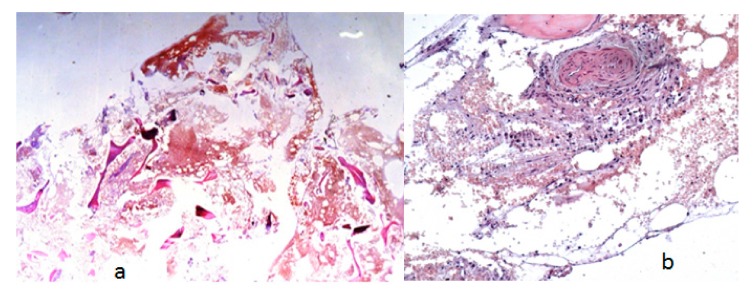
A bone specimen of a lesion over the left femur obtained at an open biopsy. Overview bone biopsy (H&E staining): (**a**) Via original magnification, the histopathology revealed trabecular fragments and large cavernous lymph spaces. (**b**) Another bone biopsy at higher magnification showed lymphatic malformation with ectatic lymph vessels. a. magnification of ×100. b. Higher magnification (×400).

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
