# Peer review of "Massive Axial and Appendicular Skeletal Deformities in Connection with Gorham-Stout Syndrome"

_medicines, 2019, doi:10.3390/medicines6020054_

Round 1

Reviewer 1 Report

The manuscript briefly describes the clinical presentation and assessment of 4 patients with a diagnosis of Gorham-Stout Syndrome.

Specific comments:

The manuscript should be edited for the English language as many sentences use improper grammar and are often not clear.

In the Introduction the authors refer to studies by Hardegger et al. (line 38 - reference #4). However, reference #4 is different and does not match that name.

Lanes 43-45: contain a repetition, i.e. the sentences state similar findings, twice.

The legend to Figure 5 also contains a repetition of the last sentence.

A small paragraph should be added to the results to report the laboratory measurements that were performed, even if these were negative.

The authors should also cite and mention a recent paper on the potential etiology of Gorham-Stout Syndrome which I think is pretty important as it highlights the potential role of lymphatic endothelial cells in the disease process.

Wang W et al. ‘Lymphatic Endothelial cells Produce M-CSF, Causing Massive Bone Loss in Mice.’ J Bone Miner Res 2017.

Author Response

The manuscript briefly describes the clinical presentation and assessment of 4 patients with a diagnosis of Gorham-Stout Syndrome.

Specific comments:

The manuscript should be edited for the English language as many sentences use improper grammar and are often not clear.

In the Introduction the authors refer to studies by Hardegger et al. (line 38 - reference #4). However, reference #4 is different and does not match that name.

Done

Lanes 43-45: contain a repetition, i.e. the sentences state similar findings, twice.

Done

The legend to Figure 5 also contains a repetition of the last sentence.

Done

A small paragraph should be added to the results to report the laboratory measurements that were performed, even if these were negative.

Done

The authors should also cite and mention a recent paper on the potential etiology of Gorham-Stout Syndrome which I think is pretty important as it highlights the potential role of lymphatic endothelial cells in the disease process.

Wang W et al. ‘Lymphatic Endothelial cells Produce M-CSF, Causing Massive Bone Loss in Mice.’ J Bone Miner Res 2017.

Done

Reviewer 2 Report

The article entitled “Massive Axial and Appendicular Skeletal Deformities In Connection with Gorham-Stout Syndrome” by Ali Al Kaissi, et al. is a case report of 4 patients with Gorham-Stout syndrome. Gorham-Stout syndrome is a rare disease characterized by destruction of osseous matrix and vascular proliferation. The authors focus on axial and appendicular skeletal deformities in this article. They concluded that It was mandatory to establish proper clinical and radiological phenotypic characterization in children and adults presented with unusual skeletal deformities. Their insistence is right, but I have concerns on the current form of the article.

Comments

1. Four cases are too small in sample size to draw a conclusion, although Gorham-Stout syndrome is an extremely rare disease. The authors are encouraged to do thorough review on skeletal deformities with Gorham-Stout syndrome to support their conclusion, especially for appendicular skeletal deformity because they show only one case in this article.

2. Additional images should be considered because the authors focus on skeletal deformity in this article.

3. A table of the presented cases and the previously reported cases will help readers to better understand axial and appendicular skeletal deformities with Gorham-Stout syndrome. 

Author Response

Dear Reviewer,

Thank you for your comments, We selected four patients, because we tried to accomplish full radiographic and tomographic documentation. Reformatted CT scan was the modality of choice to demonstrate the extent of deformity. Though, the high cost of  reformatted CT scan was the main  obstacle to include  additional patients. The four patients we choose were  models to Show the Readers the variable sites of bone destructions  regardless the Age and sex . The idea of comparing our results in a table  with the previous reports is fine, but in practice,  Gorham syndrome is notoriously unpredictable bone disease, therefore it is extremely  difficult to draw clinical or radiological lines which might lead to establish guidelines. Every single case should be clinically  assessed by its own.  For instance in patients with  syndromic associations or skeletal dysplasia, the story is totally different. Since the vast majority of such disorders manifesting frequent clinical and radiographic features which are almost constant, unless in certain cases we might encounter occasional abnormalities. But, nevertheless, the genotypic characterizations are of great help in delineating syndromic entities from a disease like Gorham.

Best regards

Al Kaissi  

 .

Reviewer 3 Report

The paper is interesting and well written.

Introduction: I only suggest to report the orphanet number of the pathology as well as the incidence of the pathology.

Author Response

Dear Editor,

We followed your comments.

Best regards

Al Kaissi

Round 2

Reviewer 2 Report

The authors have revised their article entitled “Massive Axial and Appendicular Skeletal Deformities In Connection with Gorham-Stout Syndrome” which is a case report of 4 patients with Gorham-Stout syndrome. They made a rebuttal statement, but refused to make any modifications to the main text. 

Author Response

Dear Reviewer,,

Thank you for your suggestions. We converted the paper from Article to case reports

Best regards

Al Kaissi